# Post-operative bracing following adult spine deformity surgery: Results from the AO Spine surveillance of post-operative management of patients with adult spine deformity

So Kato[1]*, Justin S. Smith[2], Devin Driesman[3], Christopher I. Shaffrey[4], Lawrence G. Lenke[5], Stephen J. Lewis[3], AO Spine Knowledge Forum Deformity[6]¶

1 Department of Orthopaedic Surgery, the University of Tokyo, Bunkyo-ku, Tokyo, Japan, 2 Department of Neurosurgery, University of Virginia Health System, Charlottesville, Virginia, United States of America, 3 Department of Orthopaedic Surgery, Toronto Western Hospital, Schroeder Arthritis Institute, University of Toronto, Toronto, Ontario, Canada, 4 Department of Orthopaedic Surgery, Duke University, Durham, North Carolina, United States of America, 5 Department of Orthopedic Spine Surgery, The Spine Hospital, Columbia University Medical Center, New York, New York, United States of America, 6 AO Foundation, Davos, Switzerland

¶ Membership of AO Spine Knowledge Forum Deformity is provided in the Acknowledgments.
* skatou-tky@umin.ac.jp

**Data Availability Statement:** All relevant data are within the manuscript.

## Abstract

### Study design

Cross-sectional international survey with a literature review.

### Objectives

While some surgeons favor spine bracing after surgery for adult spine deformity (ASD) to help prevent mechanical failures, there is a lack of evidence. The objective of the present study was to better understand the current trend in the use of bracing following ASD surgery based on an international survey.

### Methods

An e-mail-based online survey was conducted among over 6000 international AO Spine members regarding the post-operative management of patients with ASD. The details of brace prescription, indications and influencing factors were solicited. Descriptive data were summarized based on different demographic groups and fusion levels for the responding surgeons who annually perform at least 10 long-segment fusions of >5 levels extending to the pelvis.

### Results

A total of 116 responses were received, including 71 surgeons (61%) who used post-operative bracing for >5 levels of long fusion. The most common reason for bracing was pain management (55%) and bone quality was the strongest influencing factor (69%). Asia-Pacific

**Funding:** The author(s) received no specific funding for this work.

**Competing interests:** SK receives grants from Baxter and Olympus Terumo Biomaterials. JSS receives grants from DePuy Synthes/ISSG and AO Spine, royalties from Zimmer Biomet and NuVasive, consulting fees from Zimmer Biomet, NuVasive, SeaSpine, Cerapedics, and Carlsmed, has leadership roles in ISSG, and has stock or stock options for Alphatec, Nuvasive and Carlsmed. DD has no declarations. CIS receives grants from NIH and ISSG, royalties from NuVasive, Medtronic, and SI-Bone, consulting fees from Proprio, Medtronic, and SI-Bone, and has stock or stock options for Priprio and NuVasive. LGL receives grants from Scoliosis Research Society, Setting Scoliosis Straight Foundation, AO Spine NIH and ISSG, royalties from Medtronic and Acuity Surgical, consulting fees from Medtronic and Acuity Surgical. SJL has no declarations. This does not alter our adherence to PLOS ONE policies on sharing data and materials.

surgeons had the highest rate of bracing (88%), while North American surgeons had the lowest (45%). The most common type of brace used were TLSO for cases with an uppermost instrumented vertebra (UIV) in the low- or mid-thoracic spine and a cervical brace for UIV at T1-3. The majority (56%) used bracing for 6–12 weeks after surgery.

## Conclusions

The present survey demonstrated significant interest in bracing following ASD surgery, however, there is substantial variability in post-operative bracing practice. A formal study on the role of bracing in ASD surgery is needed.

## Introduction

Mechanical failures such as proximal junctional kyphosis (PJK) [1] or rod fracture following pseudarthrosis [2] are common after surgery for adult spine deformity (ASD). The reported incidence varies but PJK occurrence among long-level fusion surgery is notoriously high (30–40%) and pseudarthrosis is also as common (9–22%) [2]. Although the causes of these phenomena are multifactorial, including the patient characteristics and the nature of the mechanical construct achieved, post-operative management may also play an important role in their prevention.

Among the various instructions that surgeons could provide to their post-operative patients, brace usage has been classically and commonly proposed. Although previous surveys have indicated that many spinal surgeons prefer to prescribe bracing after fusion surgery at their discretion, [3] evidence supporting its rationale is scarce. A recent systematic review showed that only a few randomized controlled trials exist to date [4] denying benefits in terms of the rate of fusion, complications, and reoperation in lumbar degenerative cohorts [5]. Therefore, there is a lack of evidence regarding the effectiveness of bracing following long thoracolumbar fusion surgery.

Given the high frequency of mechanical failures, some surgeons may have a stronger preference for brace treatment after ASD surgery than for other types of post-operative interventions to minimize the occurrence of post-operative mechanical failures. Therefore, it is worthwhile to understand the reasons behind brace prescriptions among surgeons managing ASD, despite the lack of supporting evidence for the prevention of complications [6, 7]. The objective of the present study was to capture the current trend in surgeon preference for brace use following ASD surgery. Meanwhile, the current body of literature related to brace usage in spinal fusion surgery was reviewed for the summary.

## Methods

A peri-operative spine survey was formulated by a study group within AO Spine. The study group included experts in the knowledge forum (KF) degenerative and knowledge forum (KF) deformity spine. The questionnaire included demographic information on participants and was designed to cover various aspects of peri-operative care such as wound management, antibiotics, bracing and activity instructions. An online survey was distributed via email to AO Spine users and members between March 3 and March 22, 2022. The survey was targeted at surgeons performing at least 10 cases per year using one or more of the following procedures:

a. Long fusion (>5 levels) for adult spine deformity patients extending to pelvis

b. Long fusion (>5 levels) for adult spine deformity patients NOT extending to pelvis

c. Open 1 or 2 level fusion for adult lumbar degenerative pathologies

d. MIS 1 or 2 level fusion for adult lumbar degenerative pathologies

e. Open 3 to 5 level fusion for adult lumbar degenerative pathologies

It was estimated that over 6000 surgeons that were AO Spine users and members received the email. Among all those who received the email, 354 responded and 280 completed the survey. Of the surgeons who completed the survey, 164 performed adult spine deformity surgeries (procedures a and/or b) and 261 performed adult spinal degenerative surgeries (procedures c, d and/or e).

In the present study, focus was given to post-operative bracing practice following ASD surgery, defined as thoracolumbar fusion of > 5 levels. Questions related to bracing included 1) whether they prescribed any type of brace after surgery, 2) the main purpose for bracing, 3) factors influencing brace use, 4) duration of post-operative brace use, and 5) the type of brace prescribed depending on the surgical procedure. For questions 2) and 3), several pre-filled response options were provided along with a free-form option, allowing individuals to provide individual statements. Multiple selections were permitted. The same questions were asked for long fusion (> 5 levels) for ASD extending to the pelvis (procedure a) and not extending to the pelvis (procedure b). Among the responses obtained from 280 spinal surgeons for the entire questionnaire, 116 responses from surgeons performing these procedures were analyzed as only surgeons who performed more than 10 of these procedures per year were eligible for inclusion.

## Results

A total of 116 responses were received, including 71 surgeons (61%) who used post-operative bracing for long fusion > 5 levels, while the rest denied brace usage. Among them, 38 surgeons (33%) only occasionally prescribed the brace for selected cases, with the details of the indications unrevealed. Surgeons from the Asia-Pacific region had the highest rate of bracing (88%), while North American surgeons had the lowest (45%) (Table 1). Neurosurgeons more commonly prescribed brace treatment than orthopaedic surgeons (76% vs. 58%).

Among the 71 surgeons who prescribed braces, 34% did not use braces for fusions that did not extend to the pelvis. The most common reason for bracing was pain management (55%), construct protection (49%), PJK prevention (37%), and fusion enhancement (25%). Among the free-form responses (15%), some surgeons highlighted motives associated with patients' emotions, such as providing reassurance or serving as a reminder for precaution. Bone quality was the strongest influencing factor (69%); however, patient age (42%), procedure performed (41%), level of fusion (39%), perceived patient activity (38%), intraoperative screw purchase (34%), and patient body habitus (31%) also affected the use of bracing (Table 2).

**Table 1. Bracing practice based on geography.**

| | Response | | Total | Geographic region of spine practice | | | | |
|---|---|---|---|---|---|---|---|---|
| | | | | Europe and Southern Africa | North America | Asia Pacific | Latin America | Middle East and Northern Africa |
| | | n | 116 | 40 | 29 | 25 | 11 | 11 |
| Bracing for long fusion with or without pelvis fixation | Yes | n | 71 | 24 | 13 | 22 | 5 | 7 |
| | (Occasionally) | (n) | (38) | (14) | (7) | (10) | (4) | (3) |
| | | % | 61% | 60% | 45% | 88% | 46% | 64% |
| | Never | n | 45 | 16 | 16 | 3 | 6 | 4 |
| | | % | 39% | 40% | 55% | 12% | 55% | 36% |

**Table 2. Summary of main purposes and influencing factors for bracing in long fusion to pelvis.**

| | | | Total | Geographic region of spine practice | | | | |
|---|---|---|---|---|---|---|---|---|
| | | | | Europe and Southern Africa | North America | Asia Pacific | Latin America | Middle East and Northern Africa |
| | | n | 71 | 24 | 13 | 22 | 5 | 7 |
| Main purposes for bracing | Pain management | n | 39 | 17 | 6 | 10 | 2 | 4 |
| | | % | 55% | 71% | 46% | 46% | 40% | 57% |
| | Construct protection | n | 35 | 12 | 6 | 13 | 2 | 2 |
| | | % | 49% | 50% | 46% | 59% | 40% | 29% |
| | PJK prevention | n | 26 | 6 | 6 | 11 | 1 | 2 |
| | | % | 37% | 25% | 46% | 50% | 20% | 29% |
| | Fusion enhancement | n | 18 | 5 | 2 | 8 | 1 | 2 |
| | | % | 25% | 21% | 15% | 36% | 20% | 29% |
| | Other | n | 11 | 3 | 2 | 2 | 1 | 3 |
| | | % | 15% | 13% | 15% | 9% | 20% | 43% |
| Factors influencing brace use | Bone quality | n | 49 | 14 | 9 | 16 | 5 | 5 |
| | | % | 69% | 58% | 69% | 73% | 100% | 71% |
| | Age of patient | n | 30 | 11 | 7 | 9 | 2 | 1 |
| | | % | 42% | 46% | 54% | 41% | 40% | 14% |
| | Procedure performed | n | 29 | 12 | 6 | 7 | 1 | 3 |
| | | % | 41% | 50% | 46% | 32% | 20% | 43% |
| | Level of fusion | n | 28 | 6 | 7 | 12 | 1 | 2 |
| | | % | 39% | 25% | 54% | 55% | 20% | 29% |
| | Perceived patient activity | n | 27 | 8 | 7 | 8 | 1 | 3 |
| | | % | 38% | 33% | 54% | 36% | 20% | 43% |
| | Intraoperative screw purchase | n | 24 | 4 | 7 | 9 | 2 | 2 |
| | | % | 34% | 17% | 54% | 41% | 40% | 29% |
| | Patient body habitus | n | 22 | 8 | 5 | 7 | 2 | 0 |
| | | % | 31% | 33% | 39% | 32% | 40% | 0% |
| | Other | n | 3 | 1 | 0 | 1 | 0 | 1 |
| | | % | 4% | 4% | 0% | 5% | 0% | 14% |

Not all brace prescribers responded with valid answers to the question regarding the type of brace according to the uppermost instrumented vertebra (UIV). However, the most common type of brace used in the long fusion to the pelvis was custom-molded or prefabricated thoracolumbar-sacral orthosis (TLSO) for cases with a UIV in the low- or mid-thoracic spine (45% for UIV at T7-12, and 58% for UIV at T4-6, among valid responses), followed by hyperextension braces, such as a Jewett brace, cruciform anterior spinal hyperextension (CASH) orthosis, and dorsolumbar brace (32% for T7-12 and 22% for T4-6). For UIV at T1-3, a cervical brace, such as a sternal occiput mandibular immobilization (SOMI) brace, cervical collar, or cervico-thoraco-lumbar-sacral orthosis (CTLSO), was the most commonly used (40%) (Table 3). The majority (56%) used bracing for 6 to 12 weeks after the surgery, 7% used bracing for less than 4 weeks, and 6% used bracing for more than 20 weeks (Table 4).

## Discussion

The survey of the present study was performed to capture the current trends in brace prescription following ASD surgery, summarizing personal preferences of independent physicians working at different centers internationally. The results demonstrate significant interest in

**Table 3. Summary of brace types prescribed for long fusion to pelvis.**

| Brace type | LIV | Pelvis | | | Not extending to pelvis | | |
|---|---|---|---|---|---|---|---|
| | UIV | T7-12 | T4-6 | T1-3 | T7-12 | T4-6 | T1-3 |
| Total valid responses (n) | | 56 | 50 | 47 | 36 | 30 | 30 |
| TLSO | | 25 (45%) | 29 (58%) | 16 (34%) | 16 (44%) | 18 (60%) | 10 (33%) |
| *Custom moulded* | | 13 | 18 | 11 | 8 | 9 | 5 |
| *Prefabricated* | | 12 | 11 | 5 | 8 | 9 | 5 |
| Hyperextension brace | | 18 (32%) | 11 (22%) | 7 (15%) | 10 (28%) | 6 (20%) | 6 (20%) |
| Lumbosacral corset | | 13 (23%) | 5 (10%) | 5 (11%) | 10 (28%) | 4 (13%) | 4 (13%) |
| Cervical | | 0 (0%) | 5 (10%) | 19 (40%) | 0 (0%) | 2 (7%) | 10 (33%) |
| *SOMI* | | 0 | 2 | 7 | 0 | 1 | 4 |
| *Cervical collar* | | 0 | 2 | 11 | 0 | 0 | 5 |
| *CTLSO* | | 0 | 1 | 1 | 0 | 1 | 1 |

LIV: lower-instrumented vertebra, UIV: upper-instrumented vertebra, TLSO: thoraco-lumbar-sacral orthosis, SOMI: sternal occiput mandibular immobilization, CTLSO: cervico-thoraco-lumbar-sacral orthosis

bracing with a lack of consensus for duration, indication of bracing, and influencing factors as well as the types of brace prescribed.

In summary, while a high percentage of bracing was shown globally, significant geographical differences were observed with the Asia Pacific surgeons having a greater tendency to prescribe braces. The duration of brace wearing had a significant range, from less than 4 weeks to over 20 weeks, and the indications and influencing factors seemed to differ among the surgeons. As for brace type, most surgeons seem to aim to limit forward bending by TLSO or hyperextension braces for mid- and lower-thoracic UIV. While bracing is clearly commonly used in the post-operative management following ASD surgery, we are unable to establish

**Table 4. Summary of duration for bracing in long fusion to pelvis.**

| | Response | | Total | Geographic region of spine practice | | | | |
|---|---|---|---|---|---|---|---|---|
| | | | | Europe and Southern Africa | North America | Asia Pacific | Latin America | Middle East and Northern Africa |
| | | n | 71 | 24 | 13 | 22 | 5 | 7 |
| Duration of post-operative brace in long fusion to pelvis | < 4 weeks | n | 5 | 3 | 1 | 1 | 0 | 0 |
| | | % | 7% | 13% | 8% | 5% | 0% | 0% |
| | 4 to 6 weeks | n | 17 | 6 | 2 | 4 | 3 | 2 |
| | | % | 24% | 25% | 15% | 18% | 60% | 29% |
| | 6 to 8 weeks | n | 17 | 7 | 3 | 4 | 0 | 3 |
| | | % | 24% | 29% | 23% | 18% | 0% | 43% |
| | 8 to 12 weeks | n | 23 | 5 | 6 | 8 | 2 | 2 |
| | | % | 32% | 21% | 46% | 36% | 40% | 29% |
| | 12 to 16 weeks | n | 4 | 1 | 1 | 2 | 0 | 0 |
| | | % | 6% | 4% | 8% | 9% | 0% | 0% |
| | 16 to 20 weeks | n | 1 | 1 | 0 | 0 | 0 | 0 |
| | | % | 1% | 4% | 0% | 0% | 0% | 0% |
| | > 20 weeks | n | 4 | 1 | 0 | 3 | 0 | 0 |
| | | % | 6% | 4% | 0% | 14% | 0% | 0% |

**Table 5. A summary of previous studies investigating the effectiveness of bracing following surgeries for degenerative lumbar diseases.**

| Author | Study Design | n | Diagnosis | Operation | Fusion levels | Braces | Duration (weeks) | Outcomes | Follow-up (months) | Summarized results |
|---|---|---|---|---|---|---|---|---|---|---|
| Yee [5] | RCT | 72 | Degenerative | PLF | Not shown (≥3 in 15%) | Corset | 8 | DPQ, SF-36, fusion rate, complication | 24 | No difference |
| Soliman [8] | RCT | 43 | Degenerative | TLIF or PLF | 1 to 4 | Rigid LSO | 8 | ODI, SF-12, VAS | 3 | No difference |
| Yao [9] | RCT | 90 | Degenerative | TLIF | 1 or 2 | TLSO | 12 | ODI, VAS, fusion rate, complication | 12 | No difference |
| Ma [11] | RCT | 90 | Degenerative | MIS-TLIF | 1 or 2 | TLSO | 12 | ODI, VAS, fusion rate, complication | 12 | No difference |
| Rommel-spacher [12] | RCT | 50 | Not mentioned | PLIF | 1 or 2 | LSO | 12 | ODI, VAS | 12 | No difference |
| Fujiwara [10] | RCT | 73 | Degenerative | PLIF | 1 or 2 | LSO | 12 | JOA, JOABPEQ, RMD, VAS, fusion rate | 24 | No difference |
| Zoia [15] | RCT | 54 | LDH | Discectomy | N/A | Corset | 4 | ODI, RMD, VAS | 6 | No difference |

RCT: randomized controlled trial, PLF: posterolateral fusion, TLIF: transforaminal lumbar interbody fusion, MIS: minimally invasive surgery, PLIF: posterior lumbar interbody fusion, LSO: lumbosacral orthosis, TLSO: thoraco-lumbar-sacral orthosis, DPQ: the Dallas Pain Questionnaire, SF: short-form, ODI: Oswestry Disability Index, VAS: visual analog scale, JOA: Japanese Orthopaedic Association score, JOABPEQ: JOA-back pain evaluation questionnaire, RMD: Roland-Morris Disability Questionnaire, N/A: not applicable

guidelines or recommendations regarding post-operative brace treatment secondary to the significant heterogeneity observed in this survey.

No previous reports have specifically discussed the efficacy of bracing following ASD surgery on post-operative outcomes, while a few studies have investigated its role in other types of lumbar surgery. To date, six randomized controlled trials of lumbar fusion procedures for degenerative conditions have been reported [5, 8–12]. Yee et al. first reported a prospective randomized controlled trial to investigate the advantage of using a post-operative lumbar corset following spinal fusion in a single-center cohort of 90 patients with lumbar degenerative conditions, with 78% of them undergoing one- to two-level fusion. The corset was prescribed full-time for eight weeks, and they concluded that there were no differences between the corset and the control group with regard to outcome measures when assessing disease-specific and general health status, as well as complication rates and fusion rates at one and two years post-operatively [5]. More recently, two randomized controlled studies have been conducted. Soliman et al. focused on short-term quality of life improvement and pain relief by assessing the effect of a full-time rigid molded lumbosacral orthosis for 8 weeks [8]. Their one to four-level fusion surgery cohort did not reveal any significant differences in the Short Form (SF)-12, Oswestry Disability Index (ODI), or Visual Analog Scale (VAS) at 6 weeks and 3 months. Yao et al. confined their indication to transforaminal lumbar interbody fusion (TLIF) for one- to two-level degenerative conditions and similarly reported no benefit of full-time rigid bracing in obtaining better outcomes (VAS, ODI, fusion rates, and complications) at 1 year post-operatively [9]. Similar results were reported by the same group in a minimally invasive TLIF cohort [11]. Rommelspacher et al. also demonstrated no significant differences in ODI and VAS at 1 year after 1 or 2 level PLIF between patients wearing a lumbosacral orthosis (LSO) and those without [12]. Fujiwara et al. further showed no difference between custom-made and ready-made LSO [10]. In summary, all of previous randomized controlled trials have shown no added benefit with the use of post-operative bracing. Table 5 presents a review of the literature. One systematic review and one meta-analysis evaluated these studies as low- to moderate-quality evidence and concluded that bracing was not associated with pain reduction, fusion

enhancement, or improved patient quality of life in lumbar degenerative conditions [4, 13]. The current guidelines from the American Association of Neurological Surgery and Congress of Neurological Surgeons does not recommend bracing following instrumented fusion [14].

Numerous biomechanical studies have attempted to demonstrate the stabilizing effect of bracing. Miller et al. reported that the lumbosacral corset did not immobilize L3-S1, while the Jewett hyperextension brace and TLSO diminished mobility at L3/4 and L4/5 [16]. Subsequently, Vander Kooi et al. radiographically demonstrated the immobilizing effect of custom-molded TLSO with the potential usefulness of a thigh extender [17]. However, these biomechanical studies were designed to test mobility in healthy individuals or pre-operative patients. The impact of bracing on the mechanical load on spinal instrumentation was first investigated by Rohlmann et al [18]. They tested the load on short-segment thoracolumbar instrumentation using an implanted telemetric device in patients using a Boston brace, but there was only a negligible effect.

It is noteworthy that the evidence for these recommendations was based on findings obtained from the lumbar spine with or without short-segment fixation. Given the greater likelihood of instrumentation-related complications in long fusion surgeries, the higher tendency to prescribe post-operative external fixation following long fusion surgeries is understandable. Bogaert et al. reported the prevalence of brace prescription for lumbar fusion surgery to be 52% in their community, with the main indication being pain alleviation and improvement in fusion rate [3]. In contrast, the present survey revealed that the prescription was more favored for ASD surgery, while the rate ranged from 45–88% depending on the geography and surgeons' subspecialty. The second and third most common indications for bracing were construct protection and PJK prevention, both of which were not included in the typical reasons for prescription in lumbar degenerative conditions, reflecting surgeons' concern with regard to these common complications following ASD surgery. The survey also indicated that the presence of osteoporosis influenced bracing habits, further reinforcing the fact that there are distinctive indications for ASD surgery.

Interestingly, UIV seemed to affect the choice of brace type prescribed more than LIV. The popularity of TLSO and hyperextension braces for UIV at the mid- and low-thoracic spine, as well as the high percentage of cervical braces in UIV at the upper thoracic spine, reflected the surgeons' intention to protect the proximal end of the construct. Although biomechanical reasoning of bracing in short-segment degenerative lumbar spine conditions was not robust, there may be stronger indications for prescribing braces to prevent excessive stresses on the proximal construct at the upper and middle thoracic regions, which may be associated with a high incidence of implant loosening and PJK. As for the duration of bracing, the majority of previous studies on short fusion employed an 8 to 12 weeks full-time wearing policy post-operatively [5, 8, 9, 19]. Based on the fact that fusion rates have been generally discussed at 1- and 2-year follow-ups [4] and time to fusion in lumbar instrumentation has been reported to be approximately 1 year, [20] termination of brace wearing at 3 months seems to be theoretically premature. However, given that the patient discomfort associated with brace wearing is not negligible, [8] realistic durations have to be set, particularly for rigid braces.

The present study has several limitations inherent to the nature of the online survey. First, the response rate was relatively low unfortunately. We needed to focus on the surgeons who manage ASD on a regular basis to capture the real-world trend, which further inevitably limit the number of valid responses. Given the complexity of the questionnaire and the absence of financial rebates, we consider approximately 5% in the present survey to be reasonable. However, there was a potential risk of response bias, while the response rate alone does not determine the validity of the results [21]. Secondly, responses to questions regarding indications and influencing factors may have been potentially biased, as we provided several options for

frequently encountered responses to be chosen, while open-ended responses were also accepted in a free-form format. Thirdly, while we have made an effort to investigate the practices of surgeons who prescribe braces, it would also be meaningful to understand the rationales from those who oppose brace prescription. Braces have been associated with downsides such as cost and skin irritation.

## Conclusion

With little robust scientific evidence to support the routine use of post-operative bracing in patients with ASD, clinicians' bracing practices seem to largely depend on their experience and personal preferences. The present survey demonstrated significant interest in post-operative bracing following adult spinal deformity surgery. There is, however, substantial variability in post-operative bracing practices in terms of prescription, duration, indication, and type. A formal prospective study is warranted to determine the role of bracing following ASD surgery.

## Acknowledgments

This study was organized by AO Spine through the AO Spine Knowledge Forum Deformity, a focused group of international deformity experts. AO Spine is a clinical division of the AO Foundation, which is an independent medically-guided not-for-profit organization. Study support was provided directly through the AO Spine Research Department.

The AO Spine Knowledge Forum Deformity members active at the time of data collection, conduct of work, analysis of data and writing of the manuscript include:

Ahmet Alanay (Acibadem University School of Medicine), Alekos Theologis (University of California San Francisco), Anastasios Charalampidis (Karolinska University Hospital), André Luis (Fernandes Andújar Hospital Infantil Joana de Gusmão), Anna Rienmueller (Medizinische Universität Wien), Brett Rocos (Barts Health NHS Trust), Caglar Yilgor (Acibadem Maslak Hospital), Christopher Shaffrey (Duke University Medical Center), Christopher Ames (University of California San Diego), Christopher Nielsen (Toronto Western Hospital), Colby Oitment (McMaster University), David Polly (University of Minnesota), Eric Klineberg (University of California, Davis), Federico Landriel (Hospital Italiano de Buenos Aires), Ferran Pellisé (Spine Unit Vall d'Hebron), Ganesh Swamy (University of Calgary), Go Yoshida (Hamamatsu University School of Medicine), Jianxiong Shen (Peking Union Medical College Hospital), John Street (Vancouver General Hospital), Juan Uribe (Barrow Neurological Institute), Justin Smith (University of Virginia), Kenneth Cheung (University of Hong Kong), Kenny Kwan (The University of Hong Kong), Kristen Jones (University of Minnesota), Lawrence Lenke (Columbia University New York), Marinus de Kleuver (Radboud University Medical Center), Michael Kelly (Rady Children's Hospital), Miranda van Hooff (Radboud University Medical Center / Sint Maartenskliniek), Munish Gupta (Washington University School of Medicine), Saumyajit Basu (Kothari Medical Center and Park Clinic), Sigurd Berven (University of California San Francisco), So Kato (The University of Tokyo), Stephen Lewis (Toronto Western Hospital), Thorsten Jentzsch (Balgrist Universitätsklinik), Yong Qiu (Nanjing University), Yukihiro Matsuyama (Hamamatsu University School of Medicine), Zeeshan Sardar (Columbia University), Zezhang Zhu (Nanjing University Medical School).

Lead author: Stephen J Lewis (E-mail: Stephen.Lewis@uhn.ca).

## Author Contributions

**Conceptualization:** So Kato, Justin S. Smith, Devin Driesman, Christopher I. Shaffrey, Lawrence G. Lenke, Stephen J. Lewis.

**Data curation:** So Kato, Justin S. Smith, Christopher I. Shaffrey, Lawrence G. Lenke, Stephen J. Lewis.

**Formal analysis:** So Kato, Justin S. Smith, Christopher I. Shaffrey, Lawrence G. Lenke, Stephen J. Lewis.

**Investigation:** So Kato, Justin S. Smith, Devin Driesman, Christopher I. Shaffrey, Lawrence G. Lenke, Stephen J. Lewis.

**Methodology:** So Kato, Justin S. Smith, Devin Driesman, Christopher I. Shaffrey, Lawrence G. Lenke, Stephen J. Lewis.

**Project administration:** So Kato, Devin Driesman, Christopher I. Shaffrey, Lawrence G. Lenke, Stephen J. Lewis.

**Resources:** Devin Driesman.

**Supervision:** So Kato, Justin S. Smith, Devin Driesman, Christopher I. Shaffrey, Lawrence G. Lenke, Stephen J. Lewis.

**Validation:** Devin Driesman, Christopher I. Shaffrey, Lawrence G. Lenke, Stephen J. Lewis.

**Writing – original draft:** So Kato, Justin S. Smith, Devin Driesman, Christopher I. Shaffrey, Lawrence G. Lenke, Stephen J. Lewis.

**Writing – review & editing:** So Kato, Justin S. Smith, Devin Driesman, Christopher I. Shaffrey, Lawrence G. Lenke, Stephen J. Lewis.

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
