## [Decision Letter · Decision Letter 0]

3 Dec 2023

PONE-D-23-33208Post-operative bracing following adult spine deformity surgery: results from the AO Spine surveillance of post-operative management of patients with adult spine deformityPLOS ONE

Dear Dr. Kato,

Thank you for submitting your manuscript to PLOS ONE. After careful consideration, we feel that it has merit but does not fully meet PLOS ONE’s publication criteria as it currently stands. Therefore, we invite you to submit a revised version of the manuscript that addresses the points raised during the review process.

We look forward to receiving your revised manuscript.

Kind regards,

Mohammadreza Pourahmadi, PT, Ph.D., Postdoctoral Fellow

Academic Editor

PLOS ONE

Journal Requirements:

"I have read the journal's policy and the authors of this manuscript have the following competing interests:SK receives grants from Baxter and Olympus Terumo Biomaterials. JSS receives grants from DePuy Synthes/ISSG and AO Spine, royalties from Zimmer Biomet and NuVasive, consulting fees from Zimmer Biomet, NuVasive, SeaSpine, Cerapedics, and Carlsmed, has leadership roles in ISSG, and has stock or stock options for Alphatec, Nuvasive and Carlsmed. DD has no declarations. CIS receives grants from NIH and ISSG, royalties from NuVasive, Medtronic, and SI-Bone, consulting fees from Proprio, Medtronic, and SI-Bone, and has stock or stock options for Priprio and NuVasive. LGL receives grants from Scoliosis Research Society, Setting Scoliosis Straight Foundation, AO Spine NIH and ISSG, royalties from Medtronic and Acuity Surgical, consulting fees from Medtronic and Acuity Surgical. SJL has no declarations."

4. One of the noted authors is a group or consortium AO Spine Knowledge Forum Deformity. In addition to naming the author group, please list the individual authors and affiliations within this group in the acknowledgments section of your manuscript. Please also indicate clearly a lead author for this group along with a contact email address.

5. Please include your tables as part of your main manuscript and remove the individual files. 

Reviewers' comments:

Reviewer's Responses to Questions

**Comments to the Author**

1. Is the manuscript technically sound, and do the data support the conclusions?

Reviewer #1: Yes

Reviewer #2: No

Reviewer #3: Yes

2. Has the statistical analysis been performed appropriately and rigorously? 

Reviewer #1: N/A

Reviewer #2: I Don't Know

Reviewer #3: Yes

3. Have the authors made all data underlying the findings in their manuscript fully available?

Reviewer #1: Yes

Reviewer #2: Yes

Reviewer #3: Yes

4. Is the manuscript presented in an intelligible fashion and written in standard English?

Reviewer #1: Yes

Reviewer #2: Yes

Reviewer #3: Yes

5. Review Comments to the Author

Reviewer #1: This study was designed to understand the current trend in the use of bracing following ASD surgery. The authors conducted an e-mail-based international online survey among members in the international spine organization. Data on the details of brace prescription, indications and influencing factors were summarized. They demonstrated significant interest in post-operative bracing following ASD surgery, which largely depended on surgeons’ experience and personal preference without consensus for duration, indication of bracing, and influencing factors as well as the types of brace prescribed.

Major comment:

The authors should be commended for their efforts on this article, which required substantial effort. Although there is a lack of scientific evidence, useful information for the post-operative management of adult spinal deformity are included. Thus the actual clinical significance value of this study is high.

Reviewer #2: Thank you very much for submitting the manuscript titled "Post-operative bracing following adult spine deformity surgery: results from the AOSpine surveillance of post-operative management of patients with adult spine deformity."

Abstract:

-How many surgeons were surveyed? It would be helpful to know the response rate.

Introduction:

-What is the correlation between PJK, pseudoarthrosis and postoperative bracing for ASD surgery? To the best of my knowledge, there is none. In fact, there is no evidence supporting postoperative bracing in this context. You state as much in your second paragraph, but the interesting question is why do surgeons still prescribe bracing without evidence of efficacy?

Methods:

-280 out of 6000 is a very low response rate. And of those only 116 matched the inclusion criteria.

-For the question "the main purpose for bracing" asked in your survey, was this a free-form response? Or did you have prefilled answer choices for the survey participants?

-A question for "why do you not use a brace?" would have been interesting.

Results:

-What does it mean to use bracing selectively? What is the indication for "selective" brace use?

-Results don't appear particularly profound

Discussion:

-Should include a limitations section

Reviewer #3: This article investigates the post-operative care of patients with Adult Spinal Deformity (ASD) among spine surgeons through a mail-based online survey sent to international AO Spine members. The discussion around the necessity of wearing a back brace after ASD surgery remains an important aspect of this study. The results, which have not been published elsewhere, provide valuable insights for spine surgeons. The article is well-presented and written in standard English, making it suitable for recommendation.

6. PLOS authors have the option to publish the peer review history of their article (what does this mean?). If published, this will include your full peer review and any attached files.

Reviewer #1: No

Reviewer #2: No

Reviewer #3: **Yes: **Yu-Cheng Yao

---

## [Author Response · Author response to Decision Letter 0]

11 Dec 2023

Thank you for your kind and thoughtful comments. Please kindly see the response letter.

---

## [Decision Letter · Decision Letter 1]

9 Jan 2024

Post-operative bracing following adult spine deformity surgery: results from the AO Spine surveillance of post-operative management of patients with adult spine deformity

PONE-D-23-33208R1

Dear Dr. Kato,

We’re pleased to inform you that your manuscript has been judged scientifically suitable for publication and will be formally accepted for publication once it meets all outstanding technical requirements.

Kind regards,

Mohammadreza Pourahmadi, PT, Ph.D., Postdoctoral Fellow

Academic Editor

PLOS ONE

Additional Editor Comments (optional):

Reviewers' comments:

Reviewer's Responses to Questions

**Comments to the Author**

1. If the authors have adequately addressed your comments raised in a previous round of review and you feel that this manuscript is now acceptable for publication, you may indicate that here to bypass the “Comments to the Author” section, enter your conflict of interest statement in the “Confidential to Editor” section, and submit your "Accept" recommendation.

Reviewer #1: All comments have been addressed

Reviewer #2: All comments have been addressed

2. Is the manuscript technically sound, and do the data support the conclusions?

Reviewer #1: Yes

Reviewer #2: Partly

3. Has the statistical analysis been performed appropriately and rigorously? 

Reviewer #1: N/A

Reviewer #2: Yes

4. Have the authors made all data underlying the findings in their manuscript fully available?

Reviewer #1: Yes

Reviewer #2: Yes

5. Is the manuscript presented in an intelligible fashion and written in standard English?

Reviewer #1: Yes

Reviewer #2: Yes

6. Review Comments to the Author

Reviewer #1: This study includes enough important information in the treatment of adult spinal deformity. This article is suitable for publication in PLOS ONE.

Reviewer #2: Nice job addressing the comments. You addressed the questions of my review. Apparently there is a minimum character count that I need to exceed.

7. PLOS authors have the option to publish the peer review history of their article (what does this mean?). If published, this will include your full peer review and any attached files.

Reviewer #1: No

Reviewer #2: No

---

## [Editor Report · Acceptance letter]

1 Apr 2024

PONE-D-23-33208R1 

PLOS ONE

Dear Dr. Kato, 

I'm pleased to inform you that your manuscript has been deemed suitable for publication in PLOS ONE. Congratulations! Your manuscript is now being handed over to our production team.

Kind regards, 

on behalf of

Dr. Mohammadreza Pourahmadi 

Academic Editor

PLOS ONE